# Interacting Environmental Stress Factors Affect Metabolomics Profiles in Stored Naturally Contaminated Maize

**DOI:** 10.3390/microorganisms10050853

**Published:** 2022-04-20

**Authors:** Esther Garcia-Cela, Michael Sulyok, Carol Verheecke-Vaessen, Angel Medina, Rudolf Krska, Naresh Magan

**Affiliations:** 1Applied Mycology Group, Environment and AgriFood Theme, Cranfield University, Cranfield MK43 0AL, UK; e.garcia-cela@herts.ac.uk (E.G.-C.); c.verheecke@cranfield.ac.uk (C.V.-V.); a.medinavaya@cranfield.ac.uk (A.M.); 2Clinical, Pharmacology and Biological Sciences, School of Life and Medical Sciences, University of Hertfordshire, Hatfield AL10 9AB, UK; 3Department of Agrobiotechnology IFA-Tulln, Institute of Bioanalytics and Agro-Metabolomics, University of Natural Resources and Life Sciences Vienna, Konrad-Lorenz-Str. 20, 3430 Tulln, Austria; michael.sulyok@boku.ac.at (M.S.); rudolf.krska@boku.ac.at (R.K.); 4Institute for Global Food Security, School of Biological Sciences, Queens University Belfast, University Road, Belfast BT7 1NN, UK

**Keywords:** secondary metabolites, interacting abiotic stress, water activity, temperature

## Abstract

There is interest in understanding the relationship between naturally contaminated commodities and the potential for the production of different useful and toxic secondary metabolites (SMs). This study examined the impact of interacting abiotic stress parameters of water availability and temperature of stored naturally contaminated maize on the SM production profiles. Thus, the effect of steady-state storage water activity (a_w_; 0.80–0.95) and temperature (20–35 °C) conditions on SM production patterns in naturally contaminated maize was examined. The samples were analysed using Liquid Chromatography-Tandem Mass Spectrometry (LC-MS/MS) to evaluate (a) the total number of known SMs, (b) their concentrations, and (c) changes under two-way interacting environmental stress conditions. A total of 151 metabolites were quantified. These included those produced by species of the *Aspergillus*, *Fusarium* and *Penicillium* genera and other unspecified ones by other fungi or bacteria. There were significant differences in the numbers of SMs produced under different sets of interacting environmental conditions. The highest total number of SMs (80+) were present in maize stored at 20–25 °C and 0.95 a_w_. In addition, there was a gradation of SM production with the least number of SMs (20–30) produced under the driest conditions of 0.80 a_w_ at 20–30 °C. The only exception was at 35 °C, where different production patterns occurred. There were a total of 38 *Aspergillus*-related SMs, with most detected at >0.85 a_w_, regardless of the temperature in the 50–500 ng/g range. For *Fusarium*-related SMs, the pattern was different, with approx. 10–12 SMs detected under all a_w_ × temperature conditions with >50% produced at 500 ng/g. A total of 40–45 *Penicillium*-related SMs (50–500 ng/g) were detected in the stored maize but predominantly at 20–25 °C and 0.95 a_w_. Fewer numbers of SMs were found under marginal interacting abiotic stress storage conditions in naturally contaminated maize. There were approx. eight other known fungal SM present, predominantly in low concentrations (<50 ng/g), regardless of interacting abiotic conditions. Other unspecified SMs present consisted of <20 in low concentrations. The effect of interacting abiotic stress factors for the production of different suites of SMs to take account of the different ecological niches of fungal genera may be beneficial for identifying biotechnologically useful SMs.

## 1. Introduction

Cereals are still by far the world’s most important source of food, both for direct human consumption and, indirectly, as inputs to livestock production [1]. Under environmental stress conditions, specific fungal communities can colonise the grain and produce secondary metabolites (SMs), which may be beneficial or toxic [2,3,4]. Stored cereal grains represent heterogeneous solid substrates that have been exploited for the production of pharma-related SMs, enzymes or mycotoxins. Temperature (T) and moisture content (m.c.) are the two key abiotic factors that impact the kinetics of colonisation and SM biosynthesis in such stored solid substrates [5,6,7].

Maize is often harvested with a m.c. which is conducive to mould growth (17–19% m.c. = 0.80–0.90 water activity, a_w_) which can allow certain xerophilic or xerotolerant genera such as *Aspergillus* and *Penicillium* to colonise the substrate and produce SMs. The most important toxic SMs in maize are aflatoxins (class 1 carcinogen [8]), fumonisins and ochratoxin A. There is significant knowledge about these toxic SMs produced by species such as *Aspergillus* section *Flavi, Fusarium* section *Liseola* and *Aspergillus* section *Circumdati* species, respectively. The most important environmental factors which influence the ability of the naturally contaminating mycobiota to colonise the maize post-harvest are temperature (T) and m.c. or a_w_. A_w_ is a measure of the amount of water available for microbial growth in a substrate. These two factors interact to determine which fungi will predominantly colonise the maize resulting in different combinations of SMs being produced [9]. Fungal species such as *Aspergillus flavus* produce a range of SMs, including mycotoxins such as the aflatoxin group (AFs) and cyclopiazonic acid (CPA), and a battery of other potentially useful ones.

The production of a wide range of SMs by fungal species naturally contaminating cereals, including maize, is important to understand, especially as these may change with interacting T × a_w_ conditions during storage. This interest has grown significantly because of the development of more sophisticated analytical techniques such as Liquid Chromatography-Tandem Mass Spectrometry (LC-MS/MS) and a growing library of identifiable SMs [10]. In addition, there has been much interest in microbiomes of different food commodities and the production of different SMs, which may provide useful leads for the pharma industry or have an impact on food quality/safety [11,12]. Studies of stored cereals showed the changes which can occur in targeted metabolomics profiles under different interacting abiotic conditions [7,9]. However, in most cases, the metabolomics of solid heterogeneous substrates has been done under one set of conditions only, without including the impacts of fluxes in temperature or a_w_, which influences the predominant fungal communities colonising solid cereal-based substrates, which may simulate more natural ecological environments. Indeed, while effects of single factors such as temperature, light and pH have been examined, few if any studies have examined interacting abiotic factors to examine effects on metabolomics, especially related to fungal species for pharma-related suites of SMs [13,14,15].

A recent study correlated the production of different SMs in stored wheat with molecular approaches. This showed that toxic SMs such as aflatoxin B_1_, fumonisins, and deoxynivalenol (DON) were the most common compounds present. There was also a correlation between the presence of some SM biosynthetic genes analysed by multiplex PCR with mycotoxin detection by LC-MS/MS. However, this study predominantly considered storage of very dry wheat grain of <14.5% m.c. (≤0.70 a_w_) at which few, if any, fungi or other microorganisms can grow (Stevenson et al. 2017). Garcia-Cela et al. [9] demonstrated that naturally contaminated wheat or wheat inoculated with a specific fungal species, *Fusarium graminearum*, and stored under different interacting conditions of a_w_ × T stress changed the pattern of production of SMs significantly. This affected the amounts of metabolites present and also showed that the dominant SMs produced in stored temperate cereals were mycotoxins for which legislation exists. However, there were changes in the ratios of key metabolites, which could influence the relative contamination with individual compounds. They thus suggested that under more extreme environmental stresses, different dominant SMs may be formed, which could include both beneficial or toxic suites of compounds.

There have been few, if any, similar studies carried out with naturally contaminated maize to examine the impact of interacting abiotic factors on the spectrum of SMs and the co-occurrence of different groups of compounds which may be present [15,16,17]. There is thus a significant lack of knowledge on the impact that colonisation of such nutrient-rich solid heterogeneous substrates such as maize may have on the SM profiles produced under such interacting abiotic stress factors. These are often not considered in screening for new lead compounds for pharma or other biotechnological applications.

The objectives of this study were to examine the effect of interacting conditions of temperature (20–35 °C) × water availability (0.95–0.80 a_w_) on the changes in SM profiles, and their concentrations present due to the colonisation of a stored heterogeneous maize matrix by naturally present fungal species using LC-MS/MS. The number of SMs, their concentrations and the changes which occur under these different ecological conditions were examined.

## 2. Materials and Methods

### 2.1. Fungal Genera Present on Naturally Contaminated Maize

Serial dilution plating was used to assess the fungal contaminants naturally present on the harvested maize grain. One gram sub-samples of maize at random were placed in 9 ml of sterile water +0.01% teen 80 in 25 mL glass Universal bottles. These were shaken vigorously for 60 secs, and then a serial dilution series was made for up to 5–6 dilutions. A 0.2 mL aliquot was spread-plated of each dilution onto three replicate plates of Malt Extract Agar (MEA, CM59; Oxoid; Thermo Fisher Scientific, Hemel Hempstead, Herts, UK) and Dichloran 18% glycerol (DG18, CM0729; Oxoid; Thermo Fisher Scientific, Hemel Hempstead, Herts, UK). These represented freely available water and a lowered water activity (a_w_) media. The media were incubated at 25 °C for 7 days, and the different genera in the dilution with 5–50 colonies were enumerated. Three replicate maize samples were plated, and the means of the fungal populations of the different genera are presented as colony-forming units (CFUs) per gram sample (CFUs/g dry weight maize).

### 2.2. Maize and the Development of the Moisture Adsorption Curve

Naturally contaminated feed-grade maize grain derived from France (cv Emblem, Limagrain, France) was used in these studies. Initially, 10 g maize samples were placed in 25 mL glass Universal bottles. Known amounts of water were added to three replicates for each addition (between 0.25–1.5 mL). These were well mixed and then sealed. They were stored at 4 °C overnight for equilibration. They were then returned to 25 °C and regularly mixed until equilibration. Sub-samples of maize were used for measurement of the water activity (a_w_) of each sample using a AquaLAB Meter 4 TE (Labcell Ltd., Medstead, Hants, UK). The moisture content of the samples was also obtained by drying at 105 °C for 16 h. The data of added water against a_w_ was used to plot a water adsorption curve. This was used to calculate the accurate amounts of water to use for modifying the maize to the target a_w_ levels for the experiments.

### 2.3. Grain Storage Studies

The naturally contaminated maize grain was then modified by using the moisture adsorption curve by adding the amounts of water needed to obtain the target treatment a_w_ levels with sterile water (=0.80, 0.85, 0.90, 0.95 a_w_) in glass containers (10 g), sealed and again equilibrated at 4 °C for 24 h with periodic shaking. Each a_w_ treatment and replicates were enclosed in 16 L plastic containers containing glycerol-water solutions (2 × 500 mL) to maintain the equilibrium relative humidity (ERH) of the atmosphere the same as the maize a_w_ level to avoid changes during storage and then closed. These chambers were incubated at 20, 25, 30, and 35 °C for 11 days. For each treatment condition, three replicates were used.

### 2.4. Mycotoxins and Secondary Metabolite Analyses

*Sample preparation:* The maize samples were dried at 60 °C for 48 h, then milled and stored at 4 °C until being analysed for SMs. For this, 5 g of milled maize was extracted with 20 mL of an extraction solvent (acetonitrile/water/acetic acid 79/20/1), which was then followed by a 1 + 1 dilution using the same solvent ratio. The diluted extract (5 µL) was then directly injected into the sample port for LC-MS/MS analysis.

*LC-MS/MS parameters:* The analyses and quantification of the SMs were done with a QTrap 5500 LC-MS/MS System (Applied Biosystems, Foster City, CA, USA) that was equipped with a TurboIonSpray electrospray ionization (ESI) source and a 1290 Series HPLC System (Agilent, Waldbronn, Germany). The chromatographic separation was performed at 25 °C on a Gemini^®^ C_18_-column, 150 × 4.6 mm^2^ i.d., 5 µm particle size, equipped with a C_18_ 4 × 3 mm^2^ i.d. security guard cartridge (all from Phenomenex, Torrance, CA, USA). The details of the chromatographic method and mass spectrometric parameters have been detailed previously [18,19].

Both the positive and negative polarity modes were used for ESI-MS/MS in the time-scheduled multiple reaction monitoring (MRM). These were made in two separate chromatographic runs per sample by scanning two fragmentation reactions per analyte. The MRM detection window of each analyte was set to its expected retention time of ±27 and ±48 s in the two polarity modes (positive and negative).

LC-MS/MS parameters for compounds for which no standards were available were retrieved by performing Enhance Product Ion scans in culture extracts from the related fungal producers and verification of the product ions by comparison with literature data [20].

Quantification included the use of external calibration using serial dilutions of a multi-analyte stock solution. Results were corrected for apparent recoveries determined during method validation [10]. LOD/LOQ were determined according to the EURACHEM, involving replicate measurements of samples with a low concentration of analyte and determination of the standard deviation s0 expressed as concentration units. The LOD and LOQ are obtained by multiplying s0 with a factor of 3 and 10, respectively. The accuracy of the method has been verified on a continuous basis by regular participation in proficiency testing schemes [18,19,20].

The list of the different SMs examined in this study is presented in Table 1.

### 2.5. Statistical Analysis

Statistical analysis was performed using the package JMP^®^ Pro 13 (SAS Institute Inc., 2016. Cary, NC, USA). Datasets were tested for normality and homoscedasticity using the Shapiro–Wilk and Levene tests, respectively. In all cases, the data sets failed the normality test, despite attempts busing variable transformation to try to improve normality or homogenise the variances. The transformed data were not normally distributed, and therefore, the Wilcoxon or Kruskal–Wallis test by ranks was used for the analysis of the data. All the treatments had at least three biological replicates. The studies at 20–30 °C were carried out twice with similar types, and quantities of SMs found in the maize samples.

## 3. Results

Figure 1 shows the isolation of the populations of the predominant fungal genera from the naturally contaminated maize based on plating on two different media. This shows that *Penicillium* and *Fusarium* species were the main populations isolated, with much lower levels of the *Aspergillus* genera, including *Aspergillus* section *Flavi* and section *Nigri* and the *Aspergillus glaucus* group. Other genera occasionally were characteristic field fungi, including *Cladosporium*, *Alternaria* and some white and red yeasts.

Figure 2 shows that number of SMs found in the stored maize grain at different temperature × a_w_ conditions. The total number of SMs present was maximum at 20 and 25 °C, especially with more available water at 0.95 a_w_. At both 30 and 35 °C, fewer SMs were present and at lower concentrations. There was a significant gradual increase in the number of SMs, and the numbers present at >500 ng/g as the a_w_ was increased from 0.80–0.85 to 0.95 and all the temperatures were examined. Appendix A provides the statistical analyses of all the SMs and those produced at >500 ng/g in relation to the temperature and a_w_ conditions tested.

More detailed analyses were made of the SMs produced by different key groups of fungi in stored maize. These were predominantly related to species of *Aspergillus*, *Penicillium* and *Fusarium* genera. Figure 3 shows the numbers of SMs found related to the *Aspergillus* genus. There were 38 different SMs detected. About 15 SMs were consistently produced at ≥0.90 a_w_ at all temperatures. More specific groups of SMs were produced at higher concentrations in 0.90 and 95 a_w_ at all temperatures studied. Table 2 provides the detailed list of SMs and the actual concentrations of those found under the different temperature × a_w_ conditions for the *Aspergillus*-related species colonizing the maize. The aflatoxin-related SMs were almost all produced optimally at 30–35 °C and 0.95 a_w_. In contrast, SMs such as kojic acid and 3-nitropropionic acid were produced over the whole temperature range, especially at 0.90 and 0.95 a_w_. At 20 °C, no aflatoxin-related metabolites were produced at all the a_w_ levels examined (except for sterigmatocystin, averufin and versicolorin C at ≤2.4 µg/kg).

By far, the largest group of SMs was produced in naturally stored maize by species of the *Penicillium* genus (up to 58 SMs). Figure 4 shows the relative number of SMs detected under the different conditions, with the largest number at 20–25 °C and 0.95 a_w_. In contrast, at 35 °C, very few metabolites were produced and at very low concentrations. A detailed examination of the SMs produced by *Penicillium* species is shown in Table 3. Generally, most of the SMs, including mycophenolic acid, were predominantly produced under 20–30 °C and 0.85 and 0.95 a_w_. SMs such as festuclavin and flavoglaucin were also present. These can also sometimes also be produced by *Aspergillus* species.

A total of 27 different SMs of *Fusarium* species was found in the maize stored under different temperature × a_w_ conditions. A similar range of metabolites with different concentration groups was produced by species of this genus across all temperatures (Figure 5). The range of SMs produced by *Fusarium* species and there relative concentrations under the interacting abiotic factors is shown in Table 4. Of particular interest was the presence of both free (fumonisins, deoxynivalenol) and bound toxic SMs. Thus, hydrolysed fumonisin B_1_ and deoxynivalenol 3-glucoside were found under different interacting abiotic conditions. The *Fusarium* SMs produced optimally changed with temperature and a_w_ conditions, especially for type B trichothecenes, enniatins, 15-hydroyculmorin and chrysogin, as examples.

A few other fungal SMs were produced but in very low concentrations, regardless of the interacting abiotic conditions (Figure 6). These were generally produced in low concentrations, especially at 20–30 °C and 0.90 and 0.95 a_w_. Other unspecified SMs were found in the maize varying from 10–20 depending on the temperature and aw conditions. There was an increasing gradation of these SMs as more water was available in the substrate and at the temperature conditions examined (Figure 7).

## 4. Discussion

This study was focused on the metabolite profiles produced by different fungal genera as well as other unknown SMs. The use of different interacting storage interacting abiotic factors has shown that this can have a significant impact on the range of SMs produced and their relative concentrations. In addition, the temperature and available water influenced the predominant SMs found when this heterogeneous grain substrate and changed with different stress conditions. Indeed, the SMs produced by *Aspergillus* species was predominantly produced at >25 °C and >0.85 a_w_. In contrast, more *Penicillium* SMs were found in the maize, especially at 20–30 °C over a range of water stress conditions. *Fusarium* species, usually colonise such cereal substrates optimally under wetter conditions of >0.98 a_w_ and 25–30 °C. However, even at 0.90 and 0.95 a_w_ water stresses, there was a range of SMs produced, including both free and bound related compounds, such as the hydrolysed fumonisins and deoxynivalenol glucoside.

In relation to the discovery of novel SMs, often screening programmes have often used single temperatures on liquid/solid defined media without any modification of abiotic factors. This may limit the range of types of SMs isolated. In addition, the conditions used are unrelated to the ecological niches from which the microorganisms have been isolated. It may be more important to consider the environmental conditions which are optimum for growth vs those optimum for SM production. These are often not the same. This has been shown for the growth and production of toxic SMs such as aflatoxins, ochratoxin A and fumonisins, as well as for pharma-based products such as squalastatins and taxol [21,22]. Thus imposing duel abiotic stress factors such as temperature and water have been demonstrated to have a significant impact on the range and concentrations of SMs found in such natural heterogeneous matrices.

The SMs screened in this study and found in the maize were predominantly related to three genera which have very different resilience to temperature × water stress conditions. *Aspergillus* species are known to be xerophilic and able to grow and produce SMs over the widest water availability range. *Penicillium* species are considered to be xerotolerant or xerophilic and grow well, often under intermediate water stress and cooler temperatures. In contrast, *Fusarium* species are often mesophilic and are less tolerant of water stress, and often do not grow effectively at <0.85 a_w_ [23].

Studies with naturally contaminated stored wheat grain showed that 24 different *Fusarium* metabolites were present that could be quantified [9]. This previous study found the dominant metabolites in the wheat grain to be DON and nivalenol (NIV), then a range of enniatins (A, A1, B, B1), apicidin and Deoxynivalenol-3-glucoside at the cool temperature of 10 °C. However, increasing the temperature stimulated the biosynthesis of other SMs, including aurofusarin, moniliformin, zearalenone (ZEN) and their derivatives. When this naturally contaminated wheat was inoculated with *F. graminearum* spores, there was a significant increase in the number of SMs produced (ChisSq., *p* < 0.001). Interestingly, the relative ratios of certain groups of SMs also changed under interacting abiotic stress conditions. This approach, thus, provides different ecological niches in which different fungal species may predominate and influence the production of the suites of SMs found.

Maize represents a very nutritional sugar and lipid-rich heterogeneous matrix that can be effectively exploited by colonizing fungi producing a wide range of hydrolytic enzymes. This can be subsequently used in secondary metabolite gene clusters responsible for the key biosynthetic pathways for SM production. In general, most fungal SM compounds synthesized fall into four key chemical classes: polyketides, terpenoids, shikimic acid-derived compounds and non-ribosomal peptides [24]. Often, additional hybrid metabolites containing moieties from the different classes are produced, such as the meroterpenoids, fusions between terpenes and polyketides. The colonisation of cereals is predominantly by ascomycetes. These fungal genera often have an abundance of gene clusters of secondary metabolism than other fungal groups, including basidiomycetes, archeo-ascomycetes and chytridiomycetes. In contrast, hemi-ascomycetes and zygomycetes have none [25]. The range and types of SMs found in the maize stored under different steady-state interacting conditions influenced the predominant SMs found. Indeed, the high concentrations of SMs found (50–500 ng/g) related to *Aspergillus* (>25 SMs), *Penicillium* (>40 SMs), and *Fusarium* (10–15 SMs) genera show that physiological biosynthesis is related to enzyme groups including a range polyketide synthases (PKS), non-ribosomal protein synthases (NRPS), tryptophan synthetases (TS) and dimethylallyl tryptophan synthetases (DMATS). These are all considered to be important for the biosynthesis of SMs.

Interactions between the different species of the predominant genera inevitably occur, depending on the ecological conditions. Some of the SMs present may well be related to such interactions to provide a competitive edge in primary or secondary resource capture [26]. In contrast, there would be practically little interaction between these fungi and bacteria because the latter do not generally grow at <0.95 a_w_ [23]. Some studies have examined the effect of light, pH and temperature on Thus the type of interactions that occur will be predominantly between those fungal species which are able to grow ecologically as saprophytes or pathogens and utilize the biosynthesis of SMs for competitive advantage. It has been suggested that communication between *Aspergillus* species and bacteria can occur via the biosynthesis of suites of SMs, especially mycotoxins [27]. However, they neglected the critical role of water availability in durable commodities and the interaction with other abiotic factors which influences SM production. In most cases, the majority of bacteria are only able to grow when water is freely available (>0.99 a_w_). The complexity of interactions between Aspergillus species and the role of toxic SMs has been suggested to be very complex and determine the niches which these species fill from an ecological network point of view [28]. Indeed, maize and other heterogeneous durable commodities can be considered a man-made ecological niche in which complex interactions can occur between genera, species and the presence/absence of different pests [29].

An extensive study by Adentunji et al. [30] of maize samples from different climatic regions of Nigeria quantified the presence of up to 60 fungal targeted SM compounds. This showed that while aflatoxins and fumonisins, DON and its derivatives were dominant, *Alternaria* SMs and a range of unrelated fungal compounds were also present. The maize samples analysed were from diverse sources and not from a specifically designed study of interacting abiotic stresses on maize SMs. However, fluxes in temperature and moisture content of the maize will have influenced the rate of fungal colonization and the suites of SMs and their concentrations found.

While the importance of simulating natural ecological niches by using solid substrate systems has often been lacking, this was pointed out as being important for significantly improving the tires of different SM groups [6,7]. In addition, it was recently pointed out that many microbial biosynthetic pathways remain silent when grown on standard defined laboratory conditions and often do not result in inducing signalling and the production of potentially defence compounds [31]. Indeed they suggested that co-culture of different microbial species on solid substrates or by mixed fermentation can result in the activation of cryptic gene clusters for the induction of novel natural products. However, in such studies, no account was taken of interacting abiotic stresses and how natural colonization by mixed populations of microorganisms could influence the suites of SMs produced [32]. This also applies to microreactors which were considered an effective tool for screening and the discovery of new bioactives [33]. However, these did not try to simulate natural ecosystems. The use of solid substrate fermentation systems with heterogeneous matrices may be an effective method for the development of such screening systems, provided that the systems can remain aerobic and temperature can be effectively controlled [34]. Certainly, the use of liquid fermentation systems, especially using immobilized systems and including abiotic stresses on the functioning of gene clusters involved in SM or enzyme biosynthesis been shown to significantly increase phenotypic production [35,36,37].

## 5. Conclusions

Certainly, this type of study demonstrates the importance of considering different ecological niches and interacting abiotic stresses to simulate fluxes that occur in the environment, which will all influence the synthesis of individual and groups of SMs. The discovery of novel lead compounds should include this more ecosystem-based approach and include more heterogeneous matrices for broadening the range of SMs that may be present and enhance the opportunity for the discovery of more novel natural products.

## Figures and Tables

**Figure 1 microorganisms-10-00853-f001:**
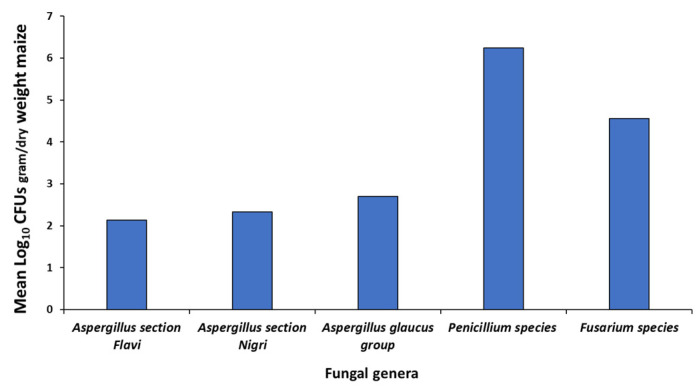
Mean fungal populations (Log_10_ CFUs/g maize) isolated from the naturally contaminated maize using serial dilution.

**Figure 2 microorganisms-10-00853-f002:**
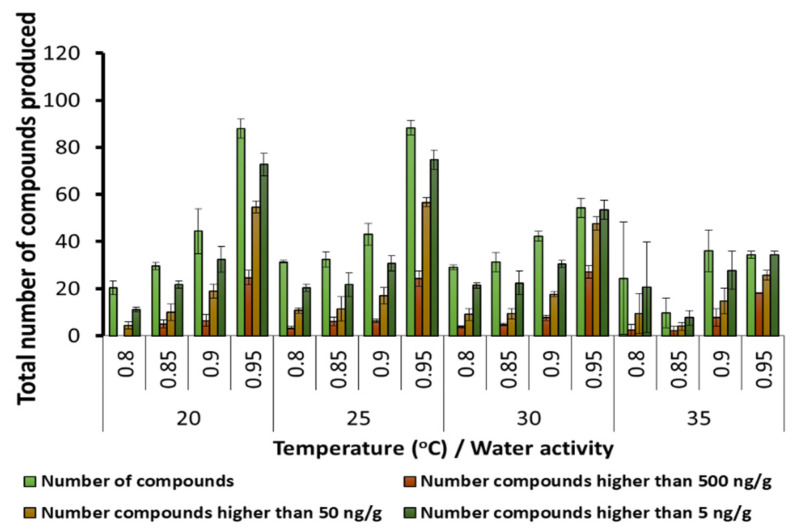
The effect of storage temperature and water activity on the total number of secondary metabolites and the number of compounds present in groupings based on relative concentrations in stored maize grain. Bars represent the Standard Error of the mean (statistics available in Appendix A).

**Figure 3 microorganisms-10-00853-f003:**
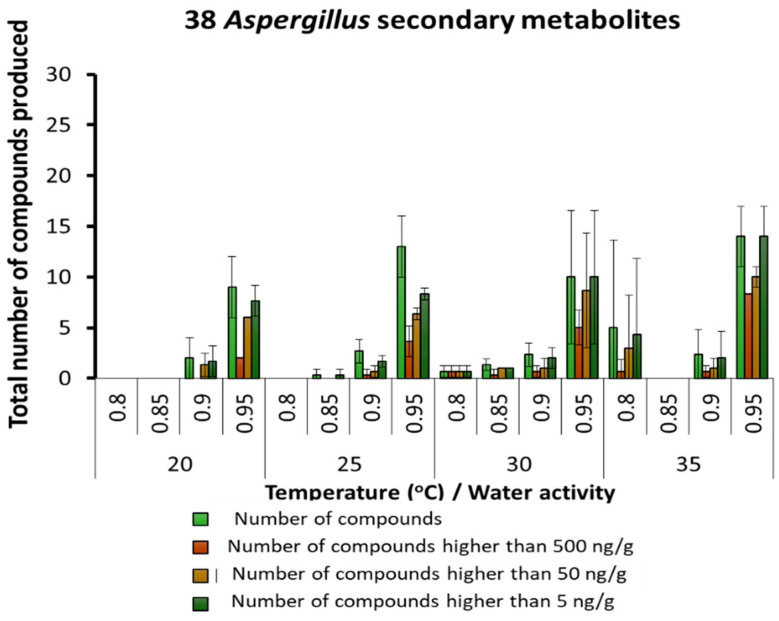
Summary of the predominant secondary metabolites (µg/g) and their concentrations detected and produced by species of the *Aspergillus* genus in maize grain stored under different temperature × water activity conditions; Bars represent the Standard Error of the mean.

**Figure 4 microorganisms-10-00853-f004:**
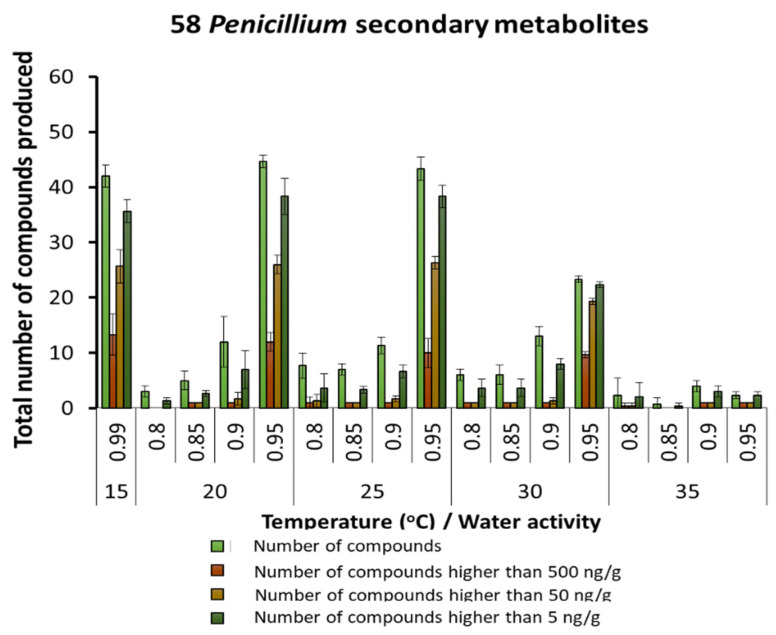
Summary of the predominant secondary metabolites detected in maize produced by species of the *Penicillium* genus stored under different temperature × water activity conditions. Bars represent the Standard Error of the mean.

**Figure 5 microorganisms-10-00853-f005:**
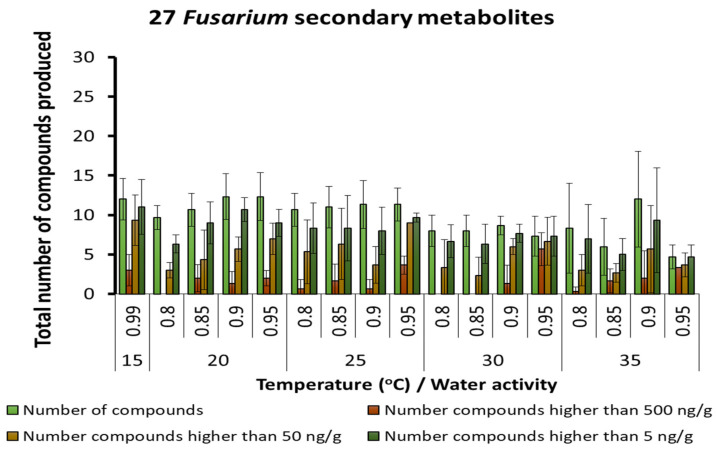
Summary of the predominant secondary metabolites of *Fusarium* species detected in maize grain stored under different temperature × water activity conditions. Bars represent the Standard Error of the mean.

**Figure 6 microorganisms-10-00853-f006:**
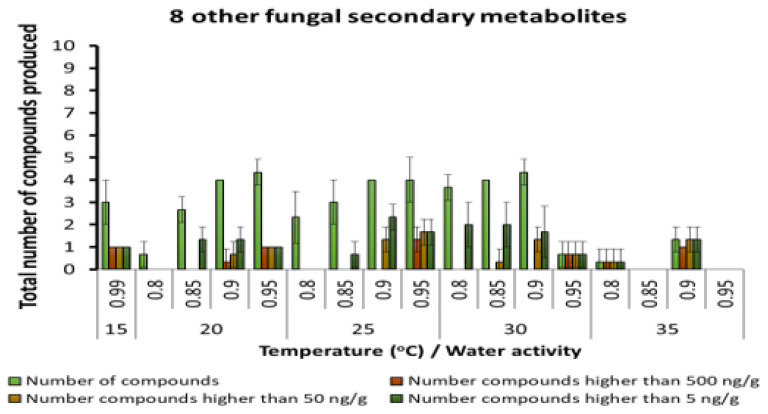
Other fungal metabolites detected in maize stored under different abiotic conditions. Bars represent the Standard Error of the mean. Note change in scale to 0–10 compounds.

**Figure 7 microorganisms-10-00853-f007:**
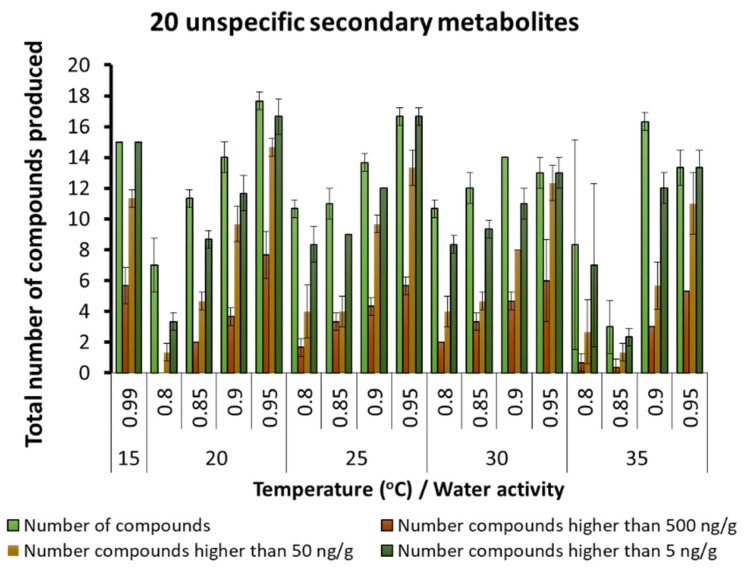
Unspecified secondary metabolites detected in the maize samples under different temperature × water activity conditions. Bars represent the Standard Error of the mean. Note change in scale to 0–20 metabolites.

**Table 1 microorganisms-10-00853-t001:** List of the different secondary metabolites examined in this study where standards were available.

Group (Number of Metabolites Studies)	Secondary Metabolites
**Aflatoxin derivatives and the metabolites from the aflatoxin pathway (14)**	Aflatoxin B_1_, Aflatoxin B_2_, Aflatoxin G_1_, Aflatoxin G_2_, Aflatoxin M_1_, Aflatoxin P_1_, Aflatoxicol. Sterigmatocystin, O-Methylsterigmatocystin, Averantin, Averufin, Versicolorin A, Versicolorin, Nidurufin, Norsolorinic acid
**Other metabolites associated with *Aspergillus* section *Flavi* species (5)**	Kojic acid, 3-Nitropropionic acid, Cyclopiazonic acid, Asperfuran, Aspinolid B
**Metabolites from other *Aspergillus* species (19)**	Aspergillicin Derivate, Aspterric acid, Bis (methylthio)gliotoxin, Butyrolacton III, Butyrolactone I, Fumigaclavine C, Fumiquinazolin D, Helvolic acid, Phenopyrrozin, Pyranonigrin, Pseurotin A, Demethylsulochrin, Methylsulochrin, Mevastatin, Viomellein, Terphenyllin, Terretonin, Dichlordiaportin, Tryprostatin B
***Fusarium* metabolites (27)**	Zearalenone, Fumonisin B_1_, Fumonisin B_2_, Fumonisin B_3_, Fumonisin B_4_, hydrolysed Fumonisin B_1_, Deoxynivalenol, DON-3-glucoside, Nivalenol, 15-Acetyldeoxynivalenol, T-2 toxin, HT-2 toxin, Monoacetoxyscirpenol, Diacetoxyscirpenol, Moniliformin, Beauvericin, Enniatin B, Enniatin B_1_, Fusarin C, Epiequisetin, Equisetin, Aurofusarin, Rubrofusarin, Bikaverin, Culmorin, 15-Hydroxyculmorin Chrysogin
***Penicillium* metabolites (58)**	Ochratoxin A, Ochratoxin B, Patulin, Citrinin, Citreoviridin, Mycophenolic acid, Mycophenolic acid IV, Norverrucosidin, Verrucosidin, Desoxyverrucosidin, Verrucofortine, Xanthomegnin, Viridicatin, Viridicatol, O-Methylviridicatin, Andrastin A, Andrastin B, Paxillin, Penicillic acid, Penicillin G, Penitrem A, Agroclavine, Chanoclavin, Festuclavine, Citreohybridinol, Oxaline, Neoxaline, Meleagrin, Pinselin, Puberulin A, Purpuride, Questiomycin A, Quinolactacin A, Roquefortine C, Roquefortine D, Rugulosin, Rugulovasine A, Griseofulvin, Griseophenone B, Griseophenone C, Dechlorogriseofulvin, Dehydrogriseofulvin, Cyclopenin, Cyclopenol, Cyclopeptine, Dehydrocyclopeptine, Flavoglaucin, Brevicompanine B, Atlantinon A, Aurantiamin A, Aurantine, Anacin, Berkedrimane B, Communesin B,2-Methylmitorubin, Pestalotin, 7-Hydroxypestalotin, Scalusamid A
**Other fungal metabolites (8)**	Altersetin, Bassianolide, Ergine, Ergometrine, Ergometrinine, Cladosporin, Gliocladic acid, Heptelidic acid
**Unspecific metabolites (20)**	Asperphenamate, Brevianamid F, Chrysophanol, Citreorosein, cyclo(L-Pro-L-Tyr), cyclo(L-Pro-L-Val), Emodin, Endocrocin, F01 1358-A, Fallacinol, Fellutanine A, Iso-Rhodoptilometrin, N-Benzoyl-Phenylalanine, Neoechinulin A, Norlichexanthone, Orsellinic acid, Physcion, Rugulusovin, Skyrin, Tryptophol

**Table 2 microorganisms-10-00853-t002:** Summary of the major secondary metabolites (ng/g) of *Aspergillus* species present in the maize under different temperature × water activity conditions. The conditions highlighted in green (low concentrations), yellow (medium concentrations) and red (highest concentrations) were found. LoD: Limit of Detection.

Maize *Aspergillus* Secondary Metabolites (ng/g)																
T (°C)	20				25				30				35			
a_w_	0.8	0.85	0.9	0.95	0.8	0.85	0.9	0.95	0.8	0.85	0.9	0.95	0.8	0.85	0.9	0.95
**Aflatoxin B_1_**	<LOD	<LOD	<LOD	<LOD	<LOD	<LOD	30.5	13.5	<LOD	1.7	<LOD	1717.2	19,732.1	<LOD	<LOD	559.2
**Aflatoxin B_2_**	<LOD	<LOD	<LOD	<LOD	<LOD	<LOD	1.7	<LOD	<LOD	<LOD	<LOD	48.5	216.9	<LOD	<LOD	22.5
**Aflatoxin G_1_**	<LOD	<LOD	<LOD	<LOD	<LOD	<LOD	<LOD	3.2	<LOD	<LOD	<LOD	<LOD	<LOD	<LOD	<LOD	<LOD
**Aflatoxin M_1_**	<LOD	<LOD	<LOD	<LOD	<LOD	<LOD	<LOD	<LOD	<LOD	<LOD	<LOD	45.5	80.6	<LOD	<LOD	18.3
**Aflatoxin P1**	<LOD	<LOD	<LOD	<LOD	<LOD	<LOD	<LOD	<LOD	<LOD	<LOD	<LOD	<LOD	1.8	<LOD	<LOD	<LOD
**Aflatoxicol**	<LOD	<LOD	<LOD	<LOD	<LOD	<LOD	<LOD	<LOD	<LOD	<LOD	<LOD	143.9	40	<LOD	<LOD	<LOD
**Sterigmatocystin**	<LOD	<LOD	<LOD	0.3	<LOD	<LOD	<LOD	1.1	<LOD	0.7	<LOD	18.4	1.1	<LOD	<LOD	<LOD
**O-Methylsterigmatocystin**	<LOD	<LOD	<LOD	<LOD	<LOD	<LOD	<LOD	0.5	<LOD	<LOD	<LOD	54.7	5.9	<LOD	<LOD	<LOD
**Averantin**	<LOD	<LOD	<LOD	<LOD	<LOD	<LOD	<LOD	1.2	<LOD	0.2	<LOD	39.3	10.5	<LOD	<LOD	27.3
**Averufin**	<LOD	<LOD	<LOD	1.1	<LOD	<LOD	<LOD	3.9	<LOD	2.2	<LOD	116	79.2	<LOD	<LOD	28.7
**Versicolorin A**	<LOD	<LOD	<LOD	<LOD	<LOD	<LOD	<LOD	4.1	<LOD	2.6	<LOD	108.2	15.1	<LOD	<LOD	25.1
**Versicolorin C**	<LOD	<LOD	<LOD	2.4	<LOD	<LOD	<LOD	5.7	<LOD	2	<LOD	158.7	277	<LOD	<LOD	68.3
**Nidurufin**	<LOD	<LOD	<LOD	<LOD	<LOD	<LOD	<LOD	0.3	<LOD	0.3	<LOD	95.8	<LOD	<LOD	<LOD	47.9
**Norsolorinic acid**	<LOD	<LOD	<LOD	<LOD	<LOD	<LOD	<LOD	3	<LOD	3	<LOD	<LOD	61.4	<LOD	<LOD	<LOD
**Kojic acid**	<LOD	<LOD	209.4	199.6	<LOD	19.5	1823.1	142,174.7	608.1	13,536.5	20,854.9	3,057,523	47,332.3	<LOD	25,061.8	3,335,509.7
**3-Nitropropionic acid**	<LOD	<LOD	158.6	499	<LOD	<LOD	<LOD	264	<LOD	182.9	56.4	13,279.1	118.1	<LOD	30.1	14,677.4
**Cyclopiazonic**	<LOD	<LOD	<LOD	<LOD	<LOD	<LOD	<LOD	137.7	552	344.8	<LOD	<LOD	<LOD	<LOD	<LOD	9244.8
**Asperfuran**	<LOD	<LOD	<LOD	<LOD	<LOD	<LOD	152.4	1212.1	<LOD	<LOD	<LOD	<LOD	389.9	<LOD	<LOD	70,155.5
**Aspinolid B**	<LOD	<LOD	0.5	244.2	<LOD	<LOD	<LOD	490.2	<LOD	355.7	<LOD	<LOD	<LOD	<LOD	<LOD	<LOD

**Table 3 microorganisms-10-00853-t003:** Detailed list and mean amounts of secondary metabolites (µg/g) found in maize related to colonization by *Penicillium* species. The conditions highlighted in green (low concentrations), yellow (medium concentrations) and red (highest concentrations) were found; LOD: Limit of Detection.

	Maize *Penicillium* Secondary Metabolites (ng/g)															
T (°C)	20				25				30				35			
a_w_	0.8	0.85	0.9	0.95	0.8	0.85	0.9	0.95	0.8	0.85	0.9	0.95	0.8	0.85	0.9	0.95
**2-Methylmitorubin**	<LOD	<LOD	<LOD	11	<LOD	<LOD	<LOD	155.1	<LOD	71.1	<LOD	413.2	<LOD	<LOD	<LOD	<LOD
**7-Hydroxypestalotin**	3.9	4.3	19.3	12.8	3.6	6.8	5	26.7	5.1	16	8.3	<LOD	9.1	2.2	<LOD	<LOD
**Agroclavine**	0.1	1.6	1.2	3.3	5.9	3	12.4	9.6	15.1	18	12.9	<LOD	<LOD	<LOD	<LOD	<LOD
**Anacin**	<LOD	<LOD	<LOD	1176.9	<LOD	<LOD	<LOD	103.7	<LOD	<LOD	<LOD	<LOD	<LOD	<LOD	<LOD	<LOD
**Andrastin A**	<LOD	0.9	1.5	166.6	<LOD	3	2.3	180.1	<LOD	270.4	4.5	1657.7	<LOD	<LOD	<LOD	<LOD
**Andrastin B**	<LOD	<LOD	<LOD	390	<LOD	<LOD	<LOD	164.7	<LOD	21.3	<LOD	133.2	<LOD	<LOD	<LOD	<LOD
**Atlantinon A**	<LOD	<LOD	7.9	729.7	<LOD	<LOD	14.2	223.2	<LOD	414.8	<LOD	<LOD	<LOD	<LOD	<LOD	<LOD
**Aurantiamin A**	<LOD	<LOD	<LOD	813.2	<LOD	<LOD	<LOD	54.4	<LOD	<LOD	<LOD	<LOD	<LOD	<LOD	<LOD	<LOD
**Aurantine**	<LOD	<LOD	2.1	969.8	<LOD	<LOD	<LOD	475.8	<LOD	1247.7	0.8	559.5	<LOD	<LOD	<LOD	<LOD
**Berkedrimane B**	<LOD	<LOD	16.8	1724.1	<LOD	<LOD	<LOD	7360.5	<LOD	3981.2	6.4	265.4	<LOD	<LOD	<LOD	<LOD
**Brevicompanine B**	<LOD	<LOD	<LOD	<LOD	<LOD	<LOD	<LOD	<LOD	<LOD	<LOD	<LOD	<LOD	<LOD	<LOD	<LOD	<LOD
**Chanoclavin**	0.3	4.1	5.3	18.6	4.1	5.6	12	17.2	8.7	9.8	6.4	24.6	<LOD	<LOD	<LOD	29.1
**Citreohybridinol**	<LOD	<LOD	<LOD	<LOD	<LOD	<LOD	<LOD	<LOD	<LOD	<LOD	<LOD	<LOD	<LOD	<LOD	<LOD	<LOD
**Citreoviridin**	<LOD	<LOD	<LOD	55.7	<LOD	<LOD	<LOD	<LOD	<LOD	<LOD	<LOD	<LOD	<LOD	<LOD	<LOD	<LOD
**Citrinin**	<LOD	<LOD	<LOD	<LOD	<LOD	<LOD	<LOD	337.9	<LOD	934.6	<LOD	<LOD	<LOD	<LOD	<LOD	<LOD
**Communesin B**	<LOD	<LOD	<LOD	19.1	<LOD	<LOD	<LOD	<LOD	<LOD	<LOD	<LOD	<LOD	<LOD	<LOD	<LOD	<LOD
**Cyclopenin**	<LOD	<LOD	<LOD	64.9	<LOD	<LOD	<LOD	107.8	<LOD	114.9	<LOD	198	<LOD	<LOD	<LOD	<LOD
**Cyclopenol**	<LOD	10.4	6.1	979.8	<LOD	<LOD	<LOD	1745.5	<LOD	1877.2	3.4	3260.8	5.1	<LOD	5.2	<LOD
**Cyclopeptine**	<LOD	<LOD	<LOD	84.1	<LOD	<LOD	<LOD	80.9	<LOD	76.9	<LOD	160.9	<LOD	<LOD	<LOD	<LOD
**Dechlorogriseofulvin**	<LOD	<LOD	<LOD	77.8	<LOD	<LOD	<LOD	301.8	<LOD	122.3	<LOD	1992.4	<LOD	<LOD	<LOD	<LOD
**Dehydrocyclopeptine**	<LOD	<LOD	<LOD	21.2	<LOD	<LOD	<LOD	19.3	<LOD	19.6	<LOD	90.1	<LOD	<LOD	<LOD	<LOD
**Dehydrogriseofulvin**	<LOD	<LOD	<LOD	0.7	<LOD	<LOD	<LOD	2.4	<LOD	0.6	<LOD	<LOD	<LOD	<LOD	<LOD	<LOD
**Desoxyverrucosidin**	<LOD	<LOD	<LOD	59.6	<LOD	<LOD	<LOD	53.5	<LOD	85.4	<LOD	118.5	<LOD	<LOD	<LOD	<LOD
**Festuclavine**	<LOD	0.2	0.6	0.9	0.2	0.3	1.7	1.5	0.5	1.1	1.6	<LOD	<LOD	<LOD	<LOD	9
**Flavoglaucin**	<LOD	7576	118,078	132,976	5186	97,681	144,820	140,188	45,908	140,928	293,943	562,099	219,519	<LOD	216,805	240,027
**Griseofulvin**	<LOD	<LOD	<LOD	83.6	<LOD	<LOD	<LOD	225.9	<LOD	139.3	<LOD	1057.6	<LOD	<LOD	<LOD	<LOD
**Griseophenone B**	<LOD	<LOD	<LOD	380.3	<LOD	<LOD	<LOD	1881.7	<LOD	644.1	<LOD	3376.9	<LOD	<LOD	<LOD	<LOD
**Griseophenone C**	<LOD	<LOD	<LOD	28	<LOD	<LOD	<LOD	126.8	<LOD	55.2	<LOD	279.5	<LOD	<LOD	<LOD	<LOD
**Meleagrin**	<LOD	<LOD	<LOD	573.9	<LOD	<LOD	<LOD	118.9	<LOD	68.6	<LOD	30.8	<LOD	<LOD	<LOD	<LOD
**Mycophenolic acid**	2.3	5.8	82.2	417.6	760	<LOD	58.3	173.2	<LOD	<LOD	<LOD	<LOD	25.7	<LOD	8.6	<LOD
**Mycophenolic acid IV**	<LOD	<LOD	2.8	9.5	1.9	<LOD	<LOD	4.1	<LOD	<LOD	<LOD	<LOD	<LOD	<LOD	<LOD	<LOD
**Neoxaline**	<LOD	<LOD	<LOD	5.9	<LOD	<LOD	<LOD	1.7	<LOD	1.6	<LOD	<LOD	<LOD	<LOD	<LOD	<LOD
**Norverrucosidin**	<LOD	<LOD	<LOD	146.2	<LOD	<LOD	1.1	84.3	<LOD	196.6	<LOD	<LOD	<LOD	<LOD	<LOD	<LOD
**Ochratoxin A**	<LOD	<LOD	<LOD	<LOD	<LOD	<LOD	<LOD	204.3	<LOD	<LOD	<LOD	<LOD	<LOD	<LOD	<LOD	<LOD
**Ochratoxin B**	<LOD	<LOD	<LOD	<LOD	<LOD	<LOD	<LOD	<LOD	<LOD	<LOD	<LOD	<LOD	<LOD	<LOD	<LOD	<LOD
**O-Methylviridicatin**	<LOD	<LOD	<LOD	8.9	<LOD	<LOD	<LOD	7.9	<LOD	11.1	<LOD	<LOD	<LOD	<LOD	<LOD	<LOD
**Oxaline**	<LOD	<LOD	0.3	293.7	0.7	<LOD	0.5	221.9	<LOD	302.2	0.1	15.8	<LOD	<LOD	<LOD	<LOD
**Patulin**	<LOD	<LOD	<LOD	<LOD	<LOD	<LOD	<LOD	<LOD	<LOD	<LOD	<LOD	<LOD	<LOD	<LOD	<LOD	<LOD
**Paxillin**	<LOD	<LOD	<LOD	<LOD	<LOD	<LOD	<LOD	<LOD	<LOD	<LOD	<LOD	<LOD	<LOD	<LOD	<LOD	<LOD
**Penicillic acid**	<LOD	<LOD	<LOD	29.9	<LOD	<LOD	<LOD	<LOD	<LOD	<LOD	<LOD	<LOD	<LOD	<LOD	<LOD	<LOD
**Penicillin G**	<LOD	<LOD	<LOD	<LOD	<LOD	<LOD	<LOD	<LOD	<LOD	<LOD	<LOD	<LOD	<LOD	<LOD	<LOD	<LOD
**Penitrem A**	<LOD	<LOD	<LOD	6.5	<LOD	<LOD	<LOD	9.1	<LOD	8.5	<LOD	<LOD	<LOD	<LOD	<LOD	<LOD
**Pestalotin**	4	<LOD	9.9	6.9	4.5	2.9	4.9	10.2	3.6	7.3	9.5	<LOD	4.7	<LOD	1.8	<LOD
**Pinselin**	<LOD	<LOD	43.5	12990.3	15.6	<LOD	37.2	3546.4	<LOD	4112.1	15.2	4634.3	<LOD	<LOD	<LOD	<LOD
**Puberulin A**	<LOD	<LOD	<LOD	94.9	<LOD	<LOD	<LOD	12	<LOD	<LOD	<LOD	<LOD	<LOD	<LOD	<LOD	<LOD
**Purpuride**	<LOD	<LOD	<LOD	1913.2	<LOD	<LOD	<LOD	3077.1	<LOD	2188.9	9.5	126.7	<LOD	<LOD	<LOD	<LOD
**Questiomycin A**	13.7	22.5	30.1	31	11.7	14.1	16.5	50.6	15.1	35.9	28.2	<LOD	20	5.9	8.7	<LOD
**Quinolactacin A**	<LOD	<LOD	<LOD	0.4	<LOD	0.1	<LOD	0.4	<LOD	0.4	51.6	2.6	<LOD	<LOD	<LOD	<LOD
**Roquefortine C**	<LOD	<LOD	<LOD	30116.3	<LOD	<LOD	<LOD	48492.9	<LOD	20269	<LOD	130454	<LOD	<LOD	<LOD	<LOD
**Roquefortine D**	<LOD	<LOD	<LOD	84.5	<LOD	<LOD	<LOD	132.8	<LOD	89	<LOD	271.9	<LOD	<LOD	<LOD	<LOD
**Rugulosin**	<LOD	<LOD	<LOD	<LOD	<LOD	<LOD	<LOD	160.1	<LOD	<LOD	<LOD	7212.7	<LOD	<LOD	<LOD	<LOD
**Rugulovasine A**	<LOD	<LOD	<LOD	28	<LOD	<LOD	<LOD	225.1	<LOD	38.9	<LOD	509.4	<LOD	<LOD	<LOD	<LOD
**Scalusamid A**	<LOD	<LOD	<LOD	<LOD	<LOD	<LOD	<LOD	<LOD	<LOD	<LOD	23.2	<LOD	<LOD	<LOD	<LOD	<LOD
**Verrucofortine**	<LOD	<LOD	<LOD	1.2	<LOD	<LOD	<LOD	0.8	<LOD	0.7	<LOD	<LOD	<LOD	<LOD	<LOD	<LOD
**Verrucosidin**	<LOD	<LOD	<LOD	335.2	<LOD	<LOD	9.9	245.5	<LOD	535.5	<LOD	125.2	<LOD	<LOD	<LOD	<LOD
**Viridicatin**	<LOD	<LOD	19.6	308.5	<LOD	<LOD	<LOD	138	<LOD	144	<LOD	407.1	<LOD	<LOD	<LOD	<LOD
**Viridicatol**	<LOD	<LOD	154.1	3473.6	204.9	<LOD	<LOD	1623.4	<LOD	1665.5	<LOD	3445.4	<LOD	<LOD	<LOD	<LOD
**Xanthomegnin**	<LOD	<LOD	<LOD	1775.7	<LOD	<LOD	<LOD	2091.7	<LOD	953	<LOD	<LOD	<LOD	<LOD	<LOD	<LOD

**Table 4 microorganisms-10-00853-t004:** Detailed list of the quantified major secondary metabolites of *Fusarium* species detected in the maize under different temperature × water activity conditions. The conditions highlighted in green (low concentrations), yellow (medium concentrations) and red (highest concentrations) were found. LOD: Limit of Detection.

	Maize *Fusarium* Secondary Metabolites (ng/g)															
T (°C)	20				25				30				35			
a_w_	0.8	0.85	0.9	0.95	0.8	0.85	0.9	0.95	0.8	0.85	0.9	0.95	0.8	0.85	0.9	0.95
**Zearalenone**	3.3	4.7	20.2	1.9	2.2	2.4	2.2	<LOD	2	1.9	23.1	<LOD	2.8	54.7	24	1303.9
**Fumonisin B_1_**	61.7	61.2	65.3	56.7	363.3	13,771.9	59.7	91,682.8	58.6	397.5	165.6	21451.5	85.9	37.1	105.5	<LOD
**Fumonisin B_2_**	19	10.5	9.7	14.7	48.3	2597.1	10.1	22,186.5	<LOD	176.2	56.4	4225.5	17.1	16.1	17.2	<LOD
**Fumonisin B_3_**	8.1	31.8	<LOD	<LOD	82.2	1011	<LOD	16,173.3	<LOD	330.9	<LOD	7480.1	31.8	<LOD	<LOD	<LOD
**Fumonisin B_4_**	<LOD	<LOD	<LOD	<LOD	96.1	1468.2	<LOD	248.2	<LOD	248.2	<LOD	5294.8	<LOD	<LOD	<LOD	<LOD
**hydrolysed Fumonisin B_1_**	<LOD	<LOD	<LOD	<LOD	<LOD	3	<LOD	78.9	<LOD	<LOD	<LOD	<LOD	<LOD	<LOD	<LOD	<LOD
**Deoxynivalenol**	339.1	1285.6	405.9	328.2	772.7	436.5	320.3	223.5	150.8	40.3	1282.6	3419.1	330.3	3213.5	537.5	6303.6
**DON-3-glucoside**	<LOD	82.9	16.4	<LOD	<LOD	<LOD	21.1	30.5	<LOD	<LOD	<LOD	<LOD	<LOD	<LOD	52.7	<LOD
**Nivalenol**	<LOD	43.6	122.1	405.9	15.9	78.6	62.2	58.9	107.6	<LOD	14.4	<LOD	161.7	51.1	1002	<LOD
**15-Acetyldeoxynivalenol**	135.4	162.4	222	81.4	94.9	188.6	111.5	<LOD	109.4	105.9	77.6	<LOD	233.5	<LOD	271.3	<LOD
**T-2 toxin**	<LOD	<LOD	3.9	<LOD	<LOD	<LOD	3.7	<LOD	<LOD	<LOD	<LOD	<LOD	22.2	<LOD	<LOD	<LOD
**HT-2 toxin**	<LOD	<LOD	30.3	<LOD	<LOD	<LOD	<LOD	<LOD	<LOD	<LOD	<LOD	<LOD	163.5	<LOD	<LOD	<LOD
**Monoacetoxyscirpenol**	<LOD	<LOD	<LOD	79.1	<LOD	<LOD	<LOD	<LOD	<LOD	<LOD	<LOD	<LOD	<LOD	<LOD	92.6	<LOD
**Diacetoxyscirpenol**	<LOD	<LOD	<LOD	<LOD	<LOD	<LOD	<LOD	<LOD	<LOD	<LOD	<LOD	<LOD	<LOD	<LOD	27.1	<LOD
**Moniliformin**	27.2	25.1	57.3	4116.3	179.7	91	26.4	2535.2	22.1	1048.2	42.1	13,282.9	19.8	24.6	447	1449
**Beauvericin**	72.8	5.8	5.8	5.1	7.2	26.8	4.7	75.9	9.7	73.6	6	171.6	5.7	7.9	9.4	10.1
**Enniatin B**	0.3	0.3	0.3	0.3	0.3	0.2	0.2	0.3	<LOD	<LOD	<LOD	<LOD	0.3	0.2	0.8	<LOD
**Enniatin B1**	0.6	0.6	0.7	0.7	0.6	0.5	0.5	0.6	<LOD	<LOD	<LOD	<LOD	0.6	0.5	0.7	<LOD
**Fusarin C**	<LOD	<LOD	<LOD	<LOD	<LOD	<LOD	<LOD	5424.2	<LOD	721.2	<LOD	15468	<LOD	<LOD	<LOD	<LOD
**Epiequisetin**	<LOD	<LOD	<LOD	<LOD	<LOD	<LOD	<LOD	<LOD	<LOD	<LOD	<LOD	<LOD	<LOD	<LOD	204.6	<LOD
**Equisetin**	<LOD	<LOD	<LOD	<LOD	<LOD	<LOD	<LOD	<LOD	<LOD	<LOD	<LOD	<LOD	<LOD	<LOD	3049.1	<LOD
**Aurofusarin**	29.7	478.9	94.8	107.1	45.8	55.8	60.5	118.8	114.4	66.2	219.3	262.1	70.4	7673.5	481.4	1929.9
**Rubrofusarin**	<LOD	<LOD	<LOD	111.7	<LOD	<LOD	<LOD	225.5	<LOD	225.5	<LOD	<LOD	<LOD	<LOD	<LOD	<LOD
**Bikaverin**	<LOD	<LOD	33.1	<LOD	120	158.1	7.5	730.4	<LOD	<LOD	<LOD	<LOD	<LOD	<LOD	<LOD	<LOD
**Culmorin**	380.8	2072.3	845.5	446.3	333	419.2	299.1	58.6	121	131.6	1794.4	4599.4	50.1	<LOD	503	<LOD
**15-Hydroxyculmorin**	<LOD	1156.1	481.9	199	472.1	205.6	89	402.1	335.7	<LOD	276.2	<LOD	723.8	12,356.5	363.5	7503.1
**Chrysogin**	<LOD	<LOD	<LOD	3.1	<LOD	<LOD	<LOD	<LOD	<LOD	<LOD	<LOD	<LOD	<LOD	<LOD	8.7	<LOD

## Data Availability

The raw research data sets are available from the corresponding author at Cranfield University.

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
