# Peer review of "Interacting Environmental Stress Factors Affect Metabolomics Profiles in Stored Naturally Contaminated Maize"

_microorganisms, 2022, doi:10.3390/microorganisms10050853_

Round 1

Reviewer 1 Report

In this work, the effect of water activity and temperature conditions on SM production patterns in naturally contaminated maize during storage was examined. By using LC-MS/MS, samples were analyzed and metabolome profiles were compared between environmental stress conditions.

Introduction section

The introduction is well-written and sets the context for the study. Among the 11 citation cited, 8 are self-citation. Work from other research teams should be presented to demonstrate a relevant literature search

Reference 11 ( Stevenson et al. 2017) is not in the right format

Materiel and methods section

Section 2.1 - Please indicate the name of the maize supplier and the variety. Is it possible to mention the initial stored maize contamination level before the storage experiments (at reception from France)?

Section 2.2- In the following sentence, one word is missing : The different aw treatments were and replicates with the same aw

Section 2.3 - Concerning the detection and quantification method, please provide LOD and LOQ in the list of the different SMs examined in the study. how are metabolites identified for which there is no commercial standard?

Section 2.4 - Experimentations and statistical analysis: Please give number of biological and technical replicates for each experiment. Have the experiments been repeated?

Please specify for which experiments the distribution of the data was not normal. Are all the data concerned?

Water activity is aw and not aw  (the w must be in index and not smaller). To be standardized throughout the document

Results section

Please standardize the units used in the figures and table (µg/kg or ng/g)

The figures are not of good quality

Discussion section

This is not really a discussion but a description of the results obtained in the study. It would be necessary to make a little more bibliography to enrich this part.

References

Among the 25 references, 15 are self-citation! this is too much and shows a narrow vision of the subject

Author Response

Referee 1

In this work, the effect of water activity and temperature conditions on SM production patterns in naturally contaminated maize during storage was examined. By using LC-MS/MS, samples were analyzed and metabolome profiles were compared between environmental stress conditions.

Introduction section

The introduction is well-written and sets the context for the study. Among the 11 citation cited, 8 are self-citation. Work from other research teams should be presented to demonstrate a relevant literature search

Reference 11 ( Stevenson et al. 2017) is not in the right format

Answer: We have now added some additional References

Materiel and methods section

Section 2.1 - Please indicate the name of the maize supplier and the variety. Is it possible to mention the initial stored maize contamination level before the storage experiments (at reception from France)?

Answer: Now added

Section 2.2- In the following sentence, one word is missing : The different aw treatments were and replicates with the same aw

Answer: has been revised.

Section 2.3 - Concerning the detection and quantification method, please provide LOD and LOQ in the list of the different SMs examined in the study. how are metabolites identified for which there is no commercial standard?

Answer: LOD/LOQ were determined according to the EURACHEM that involves replicate measurements of samples with a low concentration of analyte and determination of the standard deviation s0 expressed as concentration units. The LOD and LOQ are obtained by multiplying s0 with a factor of 3 and 10, respectively.

LC-MS/MS parameters for compounds for which no standards have been available have been retrieved by performing Enhance Product Ion scans in culture extracts from the related fungal producers and verification of the product ions by comparison with literature data (predominantly Nielsen et al. dx.doi.org/10.1021/np200254t|J. Nat. Prod. 2011, 74, 2338−48)

Section 2.4 - Experimentations and statistical analysis: Please give number of biological and technical replicates for each experiment. Have the experiments been repeated?

Answer: All data sets were found to NOT be normally distributed. We tried some transformations to normalize the data, but this did not work. Thus, the alternative approach needed to be used.

All the treatments had at least three biological replicates. The studies at 20-30oC were carried out twice with similar type, quantities of SMs found in the maize samples.

Water activity is aw and not aw (the w must be in index and not smaller). To be standardized throughout the document

Answer: I think this is wrong: aw has a subscript for the “w”. Thus Aw or aw is correct from my experience.

Results section

Please standardize the units used in the figures and table (µg/kg or ng/g)

Answer: This has now been done. Our mistake.

The figures are not of good quality

Answer: We can provide the original Figures if required.

Discussion section

This is not really a discussion but a description of the results obtained in the study. It would be necessary to make a little more bibliography to enrich this part.

Answer: We have tried to add additional text to improve this section. However, we do not feel we have repeated too many Results in the Discussion section. We have tried to highlight key interesting findings. We hope the changes made improve this section. Also added section on interactions and SMs. A number of additional References have also been added.

References

Among the 25 references, 15 are self-citation! this is too much and shows a narrow vision of the subject

Answer: We have added some more of other published work. However, it should be noted that the type of abiotic interactions examined in this study is not common in many other research studies of SMs. We have almost doubled the number of References now.

Reviewer 2 Report

In my oppinion, the topic of the manuscript is very interesting since the mycotoxins are an important food problem that are a health risk. So the identification and study of their production in different storage conditions could give new methods to avoid the production of fungal metabolites.

Some annotations are in the attached file.

Author Response

In my opinion, the manuscript deals with a very interesting topic, since it is necessary to control the mycotoxins that we find in certain foods due to the growth of fungi during the storage of fruits, cereals or vegetables. So its study in different conditions seems necessary to avoid their production. It is not clear to me how they identify the species of fungi that are being talked about, if it is because a previous isolation and identification work by molecular biology techniques or it is only because of the association with the metabolites that it produces. I think it could be clarified.

Answer: The predominant general and groups found in the maize were used as a guideline. We have now added data on the isolation of different fungal populations and the predominant groups present in the maize at the start of the experiment. We hope this gives more context.

Likewise, the metabolites that are obtained from other fungal species can also be produced by the same fungi due to interactions between them or by biotransformations of these mycotoxins from each other.

May be something to consider.

Answer: This is a good point. We have now included some text in relation to interactions and SMs and References in this aspect.

And some data about the metabolites that are not in database, that could be new mycotoxins or secondary metbolites.

Answer: The analyses used included approx. 600+ metabolites of both fungi and bacteria (Sulyok et al., 2020). Thus, we were able to examine a wide enough screen to try and identify the range of SMs produced under the different interacting abiotic factors on maize.

In conclussion, it is a complete work and that it uses appropriate techniques to identify known compounds

Answer: We believe that interacting abiotic stresses may result in some changes in SMs and ratios present. This could be important in the development of more ecologically realistic systems for the discovery of novel pharma compounds.

Round 2

Reviewer 1 Report

Thank you for taking into account the various comments that may have been made to improve the quality of the manuscript. The manuscript can be published.